# Agriculture 4.0: Polymer Hydrogels as Delivery Agents of Active Ingredients

**DOI:** 10.3390/gels10060368

**Published:** 2024-05-26

**Authors:** Aleksandra Mikhailidi, Elena Ungureanu, Bogdan-Marian Tofanica, Ovidiu C. Ungureanu, Maria E. Fortună, Dan Belosinschi, Irina Volf

**Affiliations:** 1Higher School of Printing and Media Technologies, St. Petersburg State University of Industrial Technologies and Design, 18 Bolshaya Morskaya Street, 191186 St. Petersburg, Russia; amikhailidi@gmail.com; 2“Ion Ionescu de la Brad” Iasi University of Life Sciences Iasi, 3 Mihail Sadoveanu Alley, 700490 Iasi, Romania; 3“Gheorghe Asachi” Technical University of Iasi, 73 Prof. Dr. Docent D. Mangeron Boulevard, 700050 Iasi, Romania; iwolf@tuiasi.ro; 4Faculty of Medicine, “Vasile Goldis” Western University of Arad, 94 the Boulevard of the Revolution, 310025 Arad, Romania; ungureanu.ovidiu@uvvg.ro; 5“Petru Poni” Institute of Macromolecular Chemistry, 41A Grigore Ghica Voda Alley, 700487 Iasi, Romania; fortuna.maria@icmpp.ro; 6Innovations Institute in Ecomaterials, Ecoproducts, and Ecoenergies, University of Quebec at Trois-Rivières, 3351, Boul. des Forges, Trois-Rivières QC G8Z 4M3, Canada; dbelosinschi@gmail.com

**Keywords:** bio-based hydrogels, delivery systems, fertilizer, pesticide, plant development, life cycle assessment

## Abstract

The evolution from conventional to modern agricultural practices, characterized by Agriculture 4.0 principles such as the application of innovative materials, smart water, and nutrition management, addresses the present-day challenges of food supply. In this context, polymer hydrogels have become a promising material for enhancing agricultural productivity due to their ability to retain and then release water, which can help alleviate the need for frequent irrigation in dryland environments. Furthermore, the controlled release of fertilizers by the hydrogels decreases chemical overdosing risks and the environmental impact associated with the use of agrochemicals. The potential of polymer hydrogels in sustainable agriculture and farming and their impact on soil quality is revealed by their ability to deliver nutritional and protective active ingredients. Thus, the impact of hydrogels on plant growth, development, and yield was discussed. The question of which hydrogels are more suitable for agriculture—natural or synthetic—is debatable, as both have their merits and drawbacks. An analysis of polymer hydrogel life cycles in terms of their initial material has shown the advantage of bio-based hydrogels, such as cellulose, lignin, starch, alginate, chitosan, and their derivatives and hybrids, aligning with sustainable practices and reducing dependence on non-renewable resources.

## 1. Introduction

The growing global population is driving an ever-increasing demand for agricultural products [1]. However, this is being hindered by the uneven distribution of the global population and the lack of correlation between the most populated areas and the availability of arable land suitable for cultivation in those regions. Furthermore, climate change, characterized by extreme heat, frequent droughts, and water limitations, as well as exposed and marginal soils along with erosive rain and wind, contribute to the loss of cultivable land [2].

Globalization has made food resources produced in fertile regions more accessible to all populations. International trade helps to provide the necessary amount of goods to territories of risky agriculture or those in which cultivation is impossible. However, this equilibrium is more fragile than it may seem at first glance. Political decisions can have devastating effects on regions vulnerable from a food perspective, as agricultural products and fertilizers become subject to manipulation, resulting in delayed or halted supplies, shortages, rocketing costs, and even hunger [3]. Rising fertilizer prices cause an increase in food costs that disproportionately affect poor people, emphasizing the need for policies promoting efficiency and resilience [4]. To effectively address the issue of the unequal distribution of food resources and become less reliant on the actions of politicians and businessmen, it is essential to develop novel forms of agriculture in arid regions, which cover over 40% of Earth’s terrestrial surface [5]. This pressing necessity for enhanced agricultural productivity in water-scarce areas has prompted a growing interest in innovative technologies and materials.

The third agricultural revolution, which occurred between the mid-20th century and the 1970s, was a transformative period marked by the widespread adoption of technological advancements. These included high-yielding crop varieties of cereals, chemical fertilizers and pesticides, and mechanized farming practices [6]. This revolution led to a significant increase in global food production, which addressed the challenges posed by a growing population. In order to address the issues currently facing the agricultural sector, contemporary agri-food policy has initiated a new agricultural revolution, known as Agriculture 4.0. This concept is founded on the principles of smart water and nutrition management, energy harvesting, urban farming, and the employment of digital technologies, artificial intelligence, and innovative materials [7]. The utilization of environmental biotechnology and advanced bio-based materials is in alignment with the United Nations Sustainable Development Goals and represents a significant advancement in the Industry Revolution 4.0, with the potential to facilitate a greener and more balanced future [3,8]. 

One of the valuable tools for achieving the goals of agricultural sustainability is polymer chemistry, which has demonstrated considerable success in the creation of innovative materials, such as hydrogels. A promising strategy involves encapsulating active ingredients for growth promotion, namely, fertilizers and micronutrients, as well as protective chemicals, such as herbicides, fungicides, and insecticides, which collectively are known as pesticides, within slow-release smart polymer matrices [9]. This approach offers considerable potential for addressing the prevailing challenges in agricultural development and ensuring food security.

According to ScienceDirect.com statistics, the number of articles focused on hydrogels in agriculture has tripled over the past five years, rising from 943 in 2019 to 2713 in 2023 [10]. This demonstrates a clear and unwavering interest on the part of the scientific community in this research topic. Indeed, the use of new materials simplifies plant care and increases crop yields while conserving a valuable resource, namely, water. A considerable number of reviews have been published on the subject of polymer hydrogels in agriculture [11,12,13,14,15,16,17,18,19,20]. Some of these reviews focus on specific hydrogels—such as those derived from natural polysaccharides—such as the articles by Tariq [11], Saberi Riseh [16], Qu [21], and El Idrissi [18]. Other reviews are dedicated to the application of hydrogels in niche areas of agriculture, such as urban farming [15], or explore multiple areas beyond agriculture [19]. A comprehensive review was provided by Maksimova et al., which discusses methods for obtaining polymer hydrogels of both synthetic and natural origin, with a specific emphasis on hybrid ‘semi-synthetic’ hydrogels [12]. The authors compare the properties of the three groups of polymers in order to determine the optimal option for agriculture. They conclude that each group has its own advantages and disadvantages. Another example of a detailed review is the article by Sikder et al., which differs from the previous one by focusing on the stimuli-responsiveness of polymer hydrogels, with a detailed elaboration on the mechanisms of this effect [13]. 

The present review examines polymer hydrogels from the perspective of the requirements posed by product users and the fourth industrial revolution, also known as Agriculture 4.0. The focus of this review is on the delivery of active ingredients, both nutritional and protective, and the impact of such loaded hydrogels on plant growth, development, and yield. Furthermore, attention is also given to new sustainable farming methods, such as soilless substrates, where these hydrogels have found application. In order to determine which hydrogels are more suitable for agricultural applications, we conducted a preliminary assessment of the life cycle of polymer hydrogels. It is our hope that this article will prove beneficial to researchers engaged in this and related fields, as well as to representatives of the industry involved in the production and application of polymer hydrogels.

## 2. Water Storage: A Superpower of Hydrogels

It is evident that the water utilized in conventional irrigation methods is not fully accessible to plants, with a portion being lost to evaporation. Furthermore, water and dissolved nutrients tend to migrate into deeper layers of soil, which have a low water retention capacity. This contributes to water stress conditions [19,22,23]. In light of the fact that water is scarce in many regions and that these regions are experiencing an expansion due to climate change, there is a need for the implementation of improved water management techniques in order to maintain the availability of water in the soil in proximity to plant roots.

Superabsorbent polymers are hydrogels with a three-dimensional structure that have undergone a process of cross-linking, which enables them to absorb and retain a high amount of liquid [24]. Hydrophilic superabsorbent polymers have attracted attention as a potential means of providing water reservoirs in sustainable agriculture. The incorporation of these polymers into soil can facilitate the optimization of agricultural practices, thereby mitigating the challenges posed by water scarcity. Polymer hydrogels gradually release previously captured water in the root zone due to the difference in osmotic pressure. Consequently, hydrogels facilitate the availability of approximately 95% of water to plants [22]. This characteristic significantly reduces the need for frequent irrigation, thereby ensuring optimal moisture levels for plant growth.

The swelling behavior of a hydrogel is contingent upon the degree of cross-linking, the number of hydrophilic groups, and the available space within the network structure, which is capable of accommodating, absorbing, and retaining water molecules due to its hydrophilic nature [25,26]. The swelling kinetics is a significant characteristic of hydrogels. Following precipitation or irrigation in an arid climate, rapid water evaporation occurs, accompanied by its infiltration into deeper soil strata. It is necessary for hydrogel to achieve sufficient swelling within this temporal window. 

Synthetic superabsorbent polymers have gained a preference over natural counterparts due to their superior water retention capacity [27], which can reach up to 600–1000 g/g (gram of water per gram of dry weight of the polymer) [24]. In the field of agriculture, hydrogels with water adsorption ranging from 400 to 600 g/g are commonly employed, given that the high swelling capacity of these hydrogels results in a reduction in the mechanical strength of the hydrogel [28]. Conversely, natural polymers demonstrate lower values [17]. For instance, the water retention capacity of non-derivatized cellulose hydrogels ranges from 1.7 to 66.8 g/g, depending on factors such as additives, solvent system, cross-linking agents, lignin concentration, cellulose content in the solution, pore size, temperature, and raw material [29]. Hydrogels based on chitosan exhibit comparable levels of water retention capacity [30]. For example, a chitosan hydrogel synthesized using salicylaldehyde was found to retain up to 68 g/g of water [31]. Furthermore, external factors, including pH, temperature, light, electric and magnetic fields, or the presence of specific ions, can also influence the hydrogel swelling process [32]. Despite their lower water retention capacity, bio-based hydrogels typically exhibit greater mechanical strength, particularly when reinforced with additives such as nanocrystalline cellulose [33].

Another important property of hydrogels is their capacity to release water gradually. Despite their high swelling ability, some superabsorbent polymers exhibit low water retention capacity, resulting in a rapid release. To enhance this characteristic, various fillers can be utilized, and the degree of crosslinking and monomer content can also help to control the release of water [34]. As an illustration, the incorporation of halloysite nanotubes can extend the water release duration to 30 days for a synthetic superabsorbent polymer obtained through the copolymerization of acrylamide, 2-acrylamido-2-methylpropane-1-sulfonic acid, and acrylic acid [24]. Nevertheless, the incorporation of adsorbents into the polymer composition does not invariably result in an enhanced swelling capacity. For example, the incorporation of hydrochar into carboxymethyl cellulose-g-poly(acrylic acid) hydrogel resulted in a reduction in swelling capacity of 7–21%, with the extent of this reduction varying depending on the temperature [35]. The authors demonstrated that the rate of water release is also dependent on temperature. In vivo experiments demonstrated that a hydrogel with embedded hydrochar significantly enhanced plant growth, releasing water for up to 28 days. 

Therefore, for long-term use in fields, hydrogels with a high water retention capacity should be selected, with a preference for those capable of multiple swelling cycles. Furthermore, it is important to consider the kinetics of water release, which may vary in different environmental conditions.

## 3. Active Component Delivery and Release

As novel technologies emerge and materials science advances, conventional farming methods are evolving into intelligent Agriculture 4.0 practices. This is due to the fact that materials can be tailored to achieve specific objectives. Active ingredient release systems comprise environmentally friendly formulations that are responsive to their surrounding environment, resulting in improved and regulated release characteristics [36]. A matrix delivery system involves the dissolution or uniform distribution of the active ingredient within the material, which is commonly chosen for its production simplicity. Conversely, in a reservoir delivery system, the active ingredient exists as small particles, often nanosized, encapsulated within a thin polymer shell. These systems are typically produced using microencapsulation techniques [37].

In a similar manner to their role in medicine as drug carriers for targeted therapy [38], hydrogels can be employed to deliver active ingredients to plants, commonly mineral fertilizers and pesticides. Hydrogels offer a number of advantages over other types of carriers, including films, porous beads, layered systems, and organic frameworks. This is due to the fact that the production methods for hydrogels are straightforward, which facilitates easier commercialization opportunities [39]. In contrast to synthetic polymers, which typically serve as neutral matrices for chemicals, natural polymers can act in synergy with bioactive components in the active ingredient delivery system, thereby achieving a dual effect. For instance, chitosan is not merely a carrier for active ingredients but also possesses inherent antimicrobial properties. Even in the absence of additional components, it has the potential to enhance the systemic acquired resistance of plants to pathogens and act as a protective immunostimulant and regulator of plant growth, development, and productivity [9]. 

### 3.1. Fertilizers

Following the advent of the green revolution in agriculture, global fertilizer usage increased significantly from 46.3 million tons in 1965 to 195.38 million tons in 2021, according to the Statista platform [40]. In the context of ‘precision farming’ launched by the Agriculture 4.0 concept, the focus should shift from increasing quantity to enhancing the quality of utilization. It is noteworthy that a significant proportion of nitrogen (40–70%), phosphorus (80–90%), and potassium (50–70%) from applied fertilizers is lost to the environment, resulting in pollution [41]. When essential nutrients are gradually released, it offers plants an extended duration to absorb these valuable elements. There are two types of materials: slow-release fertilizers and controlled-release fertilizers. In the case of slow-release fertilizers, a physical coating of hydrogel serves as a barrier, limiting the rate, pattern, and duration of fertilizer release (matrix type of delivery) [42]. These fertilizers begin to act after the first contact with moisture in the soil, after which the release of nutrients continues for several weeks. For example, when using a number of formulations containing carboxymethyl cellulose, acrylic acid, polyvinylpyrrolidone, and silica nanoparticles as carriers for dissolved nitrogen, phosphorus, and potassium fertilizer compounds, a gradual release was observed into both water and soil over a period of more than a month of observation [43]. 

Controlled-release fertilizers use a polymer coating for water-soluble mineral compounds (reservoir type of delivery). In response to certain stimuli, the coating may break down or become permeable, allowing the fertilizer to be released [44,45]. For instance, poly(N,N-dimethylaminoethyl methacrylate) grafted onto polydopamine-coated ammonium zinc phosphate has been observed to release nutrients in accordance with the surrounding conditions in response to changes in temperature and pH [46].

The release mechanism of fertilizer from the polymer hydrogel comprises the phases schematically illustrated in Figure 1. Initially, soil water penetrates through the coating, creating fissures as it permeates the core, which facilitates the dissolution of the fertilizer within the polymer capsule. Subsequently, as a consequence of the continued influx of water, the osmotic pressure within the core rises. This results in the collapse of the coating layer when the osmotic pressure reaches a critical threshold, which then leads to the abrupt release of the fertilizer (Figure 1, first outcome) [47,48]. Another mechanism involves the gradual release of the fertilizer through expanded pores in the polymer coating (Figure 1, second outcome). This decreases the osmotic pressure and the rate of nutrient release.

Consequently, at the initial stage, a minor proportion of the fertilizer is released, whereas, during the subsequent stage, irrespective of the mechanism, the principal portion is released. For example, the hydrogel, obtained from starch grafted with poly(acrylic acid) and loaded with urea, exhibited a similar pattern. A mere 25% of the fertilizer was released during the initial 1–5 days following its application to the soil. Subsequently, within a period of 5–20 days, over 64% of the urea was released, and in the next 20–30 days, nearly all the urea (90–99%) was loaded out [49].

In the case of controlled-release fertilizer delivery, it is typical to combine this with slow-release. Upon stimulation, the release is initiated and then occurs gradually over an extended period of time. The hydrogel, derived from cellulose nanofibres and poly(acrylamide-co-2-aminoethyl methacrylate hydrochloride), exhibited pH-responsive release properties when encapsulated with N-fertilizer. The kinetic analysis demonstrated that at pH 5.5, the release rate was 3.00 mg/day, while at pH 7.4, it decreased to 0.92 mg/day over 58 days. Similarly, at pH 5.5, the release rate was 2.69 mg/day, and at pH 7.4, it decreased to 0.55 mg/day over 65 days [50]. Thus, this release pattern encompasses both controlled and slow-release mechanisms.

Microorganisms present in soil can synthesize enzymes or acids that degrade the polymeric coating layer, thereby creating a potential pathway for the release of the fertilizer that is confined within the hydrogel shell [51]. For instance, Liang et al. proposed a 2 mm thick hydrogel coating made from waterborne polyacrylate latex for the encapsulation and subsequent controlled release of fertilizers [52]. The researchers observed that the polymer structure underwent degradation during the experiment in soil in a natural environment. They further suggested that the white-rot fungus, *Phanerochaete chrysosporium*, participated in the biodegradation. Furthermore, the infiltration of soil particles into the hydrogel film, as evidenced by the increased concentrations of C, O, K, and Si in the film composition, also contributed to the degradation process. Despite the extremely low rate of polyacrylate biodegradation (the films lost 1.77% of their weight after 12 months), after six months buried in soil, their initially smooth surfaces became rough, and numerous cracks and holes were observed. Consequently, the gradual biodegradation of synthetic hydrogels in the presence of suitable microorganisms in the soil can result in the cracking of the fertilizer reservoir and its subsequent release. However, the six-month waiting period for cracks to appear is unlikely to be a promising timeframe for real-world applications in agriculture. Conversely, it is established that environmental conditions, including temperature, pH, precipitation levels, seasonal freezing and thawing of the soil, and microbial community composition, significantly influence the rate of polymer degradation in soil [53].

It is therefore possible to hypothesize further developments in this area. Nevertheless, it appears that bio-based hydrogels are considerably more suitable for these purposes. In particular, with regard to natural polymers, the number of species of bacteria and fungi that are capable of biodegrading them is considerable [54]. Thus, the lignin-based hydrogel exhibited a loss of 6.5% after 40 days of incubation [55], while the hydrogel synthesized by graft copolymerization of acrylic acid monomers onto cassava starch exhibited a loss of 17.6% after 30 days and 72% after 120 days [56]. These results are more suitable for use in agriculture, given that they consider the typical growth cycles of plants.

### 3.2. Pesticides and Other Bioactive Ingredients

In the field of agriculture, active ingredients encompass a diverse range of substances, including fertilizers that enhance crop yields, as well as a multitude of pesticides that inhibit the growth and spread of pathogens, prevent plant diseases, and protect against insects, weeds, and other pests [57]. As previously stated, the initial stages of the agricultural revolution saw the introduction of a vast array of synthetic pesticides. As is often the case, the solutions devised by humanity to ‘improve’ nature, while effective for their intended purpose, simultaneously harm the environment by disrupting the natural balance, polluting soil and water bodies, and posing risks to the health of both people and animals. For instance, a study of acute pesticide poisoning of agricultural workers was conducted in the USA for the period from 1998 to 2005. The health of agricultural and farm employees was affected by low-severity illness (87% of total cases) and medium-severity illness (12%) associated with pesticide use. The incidence of these illnesses was found to be twice as high in women as in men. One case of poisoning resulted in a fatality [58]. In developing countries, incidents involving pesticide handlers are more prevalent, and the health consequences may manifest more rapidly due to a common absence of personal protective equipment and limited instruction on the proper application of chemical sprays. Currently, environmental policy documents advocate for the reduction of toxic chemicals in use. For instance, the Directive on the Sustainable Use of Pesticides (Directive 2009/128/EC) is designed to minimize the risks and impacts associated with pesticide use. The Federal Food, Drug and Cosmetic Act (USA) stipulates that the maximum permissible level for pesticide residues allowed in or on human food and animal feed must be set.

The polymer matrix serves to protect the active ingredients from the surrounding environment, thereby enabling a gradual and often controlled release [27]. This approach employs the same mechanisms as those used for fertilizers, thus allowing for the efficient introduction of biological control agents to suppress pests. A significant advantage of this technology is the protection of pesticides from spray drift during application, which will help to safeguard agricultural sector workers. At present, only 0.1% of pesticides reach the harmful plants and living organisms against which their action is directed [39]. The remainder is lost due to degradation or evaporation before reaching its intended target. A number of delivery systems have been proposed based on polymer hydrogels loaded with pesticides (imidacloprid, atrazine, isoproturon, carbofuran), weed growth inhibitors (uniconazole, flurpridamol, paclabutrazol, citral), genetic materials, and plant hormones (auxins) [59]. An illustrative example of a slow-release carrier is a hydrogel comprising poly(vinyl alcohol), glucomannan, and acrylamide, which was synthesized via semi-IPN (interpenetrating polymer network) formation using the gamma irradiation technique. Paraquat, a water-soluble herbicide commonly used for the control of weeds and grasses, was immobilized on the hydrogel matrix. The release of paraquat in water was gradual, with up to 36.69% of the herbicide released after 240 h of contact with water. The release rate was found to be significantly influenced by the gamma irradiation dose, with the potential for control.

As previously stated, certain hydrogels exhibit a response to stimuli in their surrounding environment, either through compression or expansion. Furthermore, plants produce protective enzymes in response to herbivorous insect attacks, which act as stimuli for the programmed release of pesticides when the plant requires pest control [39]. The objective of designing a stimuli-responsive system is to synchronize the release of nutrients from the hydrogel with a trigger linked to the plant’s requirements [47]. Stimuli-responsive materials have been investigated for the controlled release of agrochemicals in response to various factors, including pH, temperature, redox conditions, enzymes, and light [60]. Such alterations may occur in soils as a consequence of environmental fluctuations or in response to the plant’s reaction to these fluctuations [61]. An illustrative example of this concept can be observed in a biopolymeric clay hydrogel composite containing carboxymethyl cellulose, bentonite, and the insecticide thiamethoxam. The composite exhibited an initial burst release phenomenon, with a faster release rate of the active chemical observed under alkaline conditions compared to neutral conditions (pH 7.0) [62].

It is not only toxic synthetic chemicals that can be applied for plant protection; the incorporation of metal nanoparticles and/or natural biocides into hydrogel matrices represents the most effective technique currently available for the suppression of unwanted flora. For example, cellulose hydrogels containing encapsulated silver, zinc oxide, or copper (II) oxide nanoparticles have been demonstrated to effectively suppress both Gram-positive and Gram-negative pathogenic bacteria and fungi [63,64,65]. An illustrative example of natural protective compounds incorporated into a sodium alginate hydrogel is cinnamon essential oil encapsulated in silica nanoparticles. The system demonstrated remarkable efficacy against *Pseudomonas syringae*, the causal agent of pea bacterial blight, exhibiting a significant 143.58% increase in activity compared to a control. Furthermore, seeds treated with alginate and cinnamon exhibited accelerated germination rates compared to control plants, thereby underscoring the efficacy of the treatment [66]. In their study, Clemente and colleagues employed bioactive plant secondary metabolites (glycoalkaloids) extracted from agricultural waste. These compounds are naturally produced by plants as a means of protecting themselves from pathogens. The delivery carriers were crosslinked hydrogel beads produced from sodium alginate and sodium carboxymethyl cellulose. The samples demonstrated a significant reduction in the overall fungal load in agricultural soil compared to the untreated control soil, with a reduction observed as early as two weeks after the implementation of hydrogel beads containing glycoalkaloids [59].

Hydrogels can be employed for dual purposes by incorporating multiple active ingredients simultaneously for a variety of applications. For example, a polyvinyl alcohol/polyvinylpyrrolidone hydrogel was loaded with ‘green’ pesticides, namely, hydrophilic hydrogen peroxide and hydrophobic essential oil thymol, while urea was incorporated into the composition as a fertilizer [67]. Hydrophilic substances are retained by hydrogen bond interactions with the polymer structure of hydrogels. The essential oil interacts with polyvinylpyrrolidone in a hydrophobic manner due to its terpene side, while its phenolic hydrophilic side can also form hydrogen bonds. Furthermore, the simultaneous loading of multiple substances may result in interactions between them, as exemplified by the case of urea and hydrogen peroxide, thymol, and urea [67].

The delivery of active ingredients to a plant can be achieved in a number of ways. Once a hydrogel has been saturated with water and the active ingredients stored within have been dissolved, they are released upon direct contact with the roots, foliage, crop shells, or pests. The application of the hydrogel to the leaves of the plant is achieved through the spraying of a precursor solution, which forms a thin film of transparent hydrogel. This then solidifies directly on the leaves and stems of the plant. Another method involves the placement of the hydrogel in proximity to its target with a controlled air gap (e.g., beads, pots, granules), which allows for the gradual release of pesticides or moisture vapor over time, thereby creating a localized environment around the plant. Nevertheless, this approach may result in insufficient doses and some loss. Consequently, the selection of the optimal method is contingent upon the resilience of the plant and the efficacy of the released substances [67]. 

### 3.3. Limitations

Despite the apparent attractiveness of active compound delivery in the context of environmental concerns associated with the overuse of mineral fertilizers and pesticides, there are several drawbacks of these materials. 

Firstly, there is a lack of standardized methods for accurately determining the active compound release rate, which is an important consideration [48]. 

Secondly, there is a lack of correlation between data obtained through laboratory studies and the actual nutrient release rate in practical applications. It is important to note that the time of fertilizer release in water and soil varies significantly [68], and this should be taken into account when developing experimental designs. For instance, the double-network hydrogel, derived from sodium alginate, was created through the polymerization of acrylic acid, acrylamide, and β-cyclodextrin in the presence of urea-loaded halloysites. This hydrogel exhibited a relatively rapid release in water, with 87.8% released within four hours. In contrast, the release process in soil took several days, with 79.5% released within four days [69].

Thirdly, the encapsulation of pesticides and nutrients minimizes losses during release. However, excessive doses released directly onto the roots and other parts of the plant can cause phytotoxic effects and harm the plant [67].

Finally, the production of hydrogels loaded with active ingredients is a significant financial burden. The widespread adoption of this technology is significantly hindered by the higher production costs, which range from 10 to 30 times those of conventional fertilizers. This is due to the high prices of coating materials and inefficiencies in coating practices [45]. Nevertheless, enhancements to the coating processes, the effective scaling of production, and the development of regeneration technologies for reagents will assist in overcoming these limitations. 

## 4. Effects of Polymer Hydrogels on Growing Media

### 4.1. Impact on the Soil Structure 

The application of hydrogels confers a number of advantages, including the enhancement of soil structure, which, in turn, leads to improved fertility. The hydrogel undergoes repeated swelling and shrinkage processes, which positively influence the permeability, bulk density, texture, and aeration of the soil [70,71]. The hydrogels facilitate the development of microcapillaries by mechanically obstructing larger tubules in the soil. This process restricts evaporation, gravitational water outflow, and the loss of dissolved nutrients [72]. The application of hydrogels has been demonstrated to increase the number of soil aggregates with dimensions of 0.25–10 mm, which are characteristic of fertile soil. Alterations in the soil structure result in a reduction in the infiltration rate, a decrease in the presence of suspensions in soil leachate, and, ultimately, a reduction in erosion [73]. 

Synthetic polymer conditioners, such as polyacrylamide, have been demonstrated to effectively enhance soil stability, with a reduction in soil erosion of 72% observed on slopes (the aforementioned figure applies to slopes) [74]. However, since the improvements in soil quality are achieved through the application of 1–20 kg of polyacrylamide per hectare [75], the total cost of this method is rather high [76]. Another example of the application of a synthetic polymer, polyvinyl alcohol (PVA), as a soil conditioner was reported by Aly [31]. The use of PVA at concentrations of 0.1–0.2% was found to reduce the depletion of available soil moisture in sandy soil during field experiments conducted in the semi-arid climate of Egypt. This resulted in a significant improvement in faba bean yield (833.53 and 1118.78 kg/fed for untreated soil and soil with PVA conditioner, respectively) and water use efficiency (0.38 and 0.58 kg/m^3^, respectively) during two seasons compared to the control [77]. Hydrogels based on natural polymers also permit the prolonged retention of moisture in the soil, thereby preventing its evaporation and loss to deeper layers. For example, a chitosan hydrogel has been demonstrated to enhance the water retention capacity of soil by up to 154% in comparison to a control without hydrogel [31]. 

The hydrogels not only contribute to the improvement of the hydro-physical properties of the soil but also bring about beneficial changes in the biochemical properties and nutritional status of the soil. These changes ultimately influence plant growth and yields. For instance, the utilization of hydrogels derived from partially cyanoethylated rice straw and acrylonitrile had a favorable impact on the biochemical properties of the soil. This included a slight reduction in soil pH, an enhancement in cation exchange capacity, an increase in organic matter (C, N) content, and an improvement in the biological activity of the soil, along with the activities of dehydrogenase and phosphatase [78].

Dorraji et al. demonstrated that the use of a hydrophilic hydrogel, Superab A200, can mitigate the adverse impact of soil salinity on plants. The addition of 0.6% w/w of the polymer at a salinity level of 4 ms/cm resulted in an increase in available water content by 2.2 and 1.2 times more than that of controls in loamy sand and sandy clay loam soils, respectively [79]. 

In conclusion, the utilization of hydrogels has a beneficial impact on soil structure, resulting in enhanced fertility due to the influence on various soil properties and a reduction in erosion. As outlined in Grabowska-Polanowska’s review, natural polymers exhibit a lesser improvement in infiltration and surface runoff compared to synthetic ones, necessitating their application in larger quantities [73].

### 4.2. Biodegradation

Polymer hydrogels are designed to retain soil moisture and provide a gradual supply of water to plants while also biodegrading in the soil without causing environmental pollution. Hydrogels based on cellulose, starch, lignin, chitosan, gelatin, and other natural materials degrade readily in the presence of soil microorganisms, resulting in a reduced environmental impact [29,73,80]. For example, cellulase, an enzyme that breaks down cellulose, is secreted by numerous bacteria and fungi, including aerobic bacteria. The following bacteria have been identified as capable of degrading cellulose: *Bacillus subtilis*, *Cellvibrio gilvus*, and *Klebsiella pneumonia*. Anaerobic bacteria have also been observed to degrade cellulose. *Clostridium acetobutylium* and *Fibrobacter succinogenes* are examples of bacteria that are capable of utilizing cellulose. Similarly, microfungi, including *Aspergillus aculeatus*, *Chrysosporium pannorum*, *Bispora betulina*, and *actinomycetes*, are also able to degrade cellulose. The following species have been identified: *Cellulomonas biazotea*, *Micromonospora chalcea*, and *Thermomonas curvata*. A total of 60 species have been reported in the literature [81]. Furthermore, some of these bacteria, for instance, *Bacillus subtilis*, are capable of performing multiple functions. They secrete not only cellulase but also other enzymes, such as amylase, which is responsible for the degradation of starch [82]. The action of cellulase results in a reduction in the molecular weight and mechanical strength of cellulose while increasing its solubility, leading to degradation [72,83]. For example, cellulose hydrogels derived from okara were completely degraded within 28 days in soil [84]. From the perspective of agricultural use, where the vegetation period of plants often extends for a longer duration, a degradation period of less than one month may be disadvantageous. Nevertheless, such hydrogels can be utilized for, for instance, the application of fertilizer solutions, which is of particular importance during the active growth period. An alternative approach may involve the utilization of rapidly degradable cellulose hydrogels for the cultivation of greens and microgreens, whose vegetation period is limited to a few weeks.

Another example considers the impact of additives, which can extend the period of degradation to a more desirable length [85]. Cellulose hydrogels containing 4% cellulose nanocrystals (CNC) exhibited a 97% mass loss after 10 days of incubation in an extraction solution of soil. However, as the CNC content increased to 5% and 6%, the biodegradability significantly decreased to 40% and 35%, respectively. This decline was attributed to the elevated crosslinking and reduced pore size, which hindered the infiltration of soil microorganisms into the swollen hydrogel network. Consequently, by modifying the composition of a naturally derived hydrogel or merely adjusting the degree of crosslinking, it is possible to optimize the degradation period of the hydrogel for a specific plant culture. It has been suggested that hydrogels derived from polysaccharides are entirely safe supplements for soil [73].

It is also the case that hybrid synthetic polymer materials can undergo biodegradation [80,86]. Nevertheless, only a limited number of microbial species are capable of tolerating the complex structures of these materials. To mitigate the inhibitory effects of such polymers on microorganisms, they can be functionalized and/or combined with other compounds that are more environmentally friendly. For instance, although polysiloxanes are renowned for their non-biodegradability, copolymers of ε-caprolactone with aminopropyl polymethylsiloxane have been demonstrated to be susceptible to two types of micromycetes: *Fusidium viride* and *Penicillium brevicompactum* [86]. The combination of synthetic and natural polymers is proposed as a means of enhancing their biodegradability. One example is a composite of arabinogalactan with low-density polyethylene, which, according to researchers, imparts biodegradability to such materials [87]. Finally, some synthetic polymers undergo biodegradation without modification. For example, *Aspergillus fumigatus* has been reported to be responsible for the biodegradation of a range of polymers, including polycaprolactone, polyhydroxybutyrate, poly(lactic acid), and poly(1,4-butylene) succinate, at temperatures between 25 and 37 °C [88]. 

The rate of hydrogel decomposition is contingent upon the structure and composition of the hydrogel. Synthetic polymers, such as polyacrylamide and poly(acrylic acid), demonstrate limited degradability in soil and resistance to microbial degradation. The decomposition process typically takes years, with crosslinked polymers experiencing a decomposition rate of no more than 10% per year [73,89]. Furthermore, it has been demonstrated that the decomposition of polyacrylamide can result in the generation of carbonyl compounds and carboxylic acids [76], as well as the formation of undesirable quantities of acrylamide, acrylic acid, and poly(acrylic acid) [89,90]. These molecules are typically more hydrophilic, which renders them more susceptible to entering groundwater and poses a risk of contamination [89].

One approach to addressing the non-biodegradability of synthetic polymers is to combine them with natural polymers, resulting in hybrid hydrogels [12,49,91,92]. In this approach, the advantages inherent in synthetic polymer hydrogels, such as high adsorption capacity, cost-effectiveness, and the ability for multiple swelling and drying cycles, are combined with the renewable and biodegradable nature of natural polymers. A number of studies have described the combination of natural polymers crosslinked with acrylates as biodegradable composites [49,91,92]. For instance, Sarmah and Karak reported the fabrication of a superabsorbent hydrogel (water adsorption capacity up to 700 g/g) from starch-modified poly(acrylic acid) with ratios of 1:0.78, 1:1.05, and 1:2.01 wt/wt, respectively [49]. The biodegradability of the hydrogel was evaluated in vitro using two bacterial strains, namely, *Pseudomonas aeruginosa* (Gram-negative) and *Bacillus subtilis* (Gram-positive). The authors observed that the hydrogel network underwent significant degradation, with the formation of very small fragments, after six weeks of incubation. The extent of degradation was quantified by the proportion of starch present, with a higher concentration resulting in a more pronounced mass loss over time. For comparison, a hydrogel consisting solely of poly(acrylic acid) without starch additives exhibited considerably lower degradation by microorganisms (60% vs. 10%, respectively) [49]. It is of concern that the authors reported that the hydrogels broke down into small fragments at the conclusion of the experiment. The fate of these fragments raises concerns for the future. Furthermore, the experimental conditions, which involved incubation in an optimal nutrient medium for microbial proliferation, differed significantly from the reality of soil ecosystems. Indeed, the hydrogels containing starch and poly(acrylic acid) exhibited a weight loss of approximately 25–40% within three months in soil, while the poly(acrylic acid) hydrogel demonstrated a weight loss of less than 5% during the same period of exposure [49].

### 4.3. Soilless Cultivation Technologies

The practice of cultivating plants without soil in a controlled environment is a modern agricultural technology known as Controlled Environment Agriculture. This technology is predominantly implemented in industrial greenhouses. The evolution of well-suited growing media, which may be organic or inorganic materials that provide anchorage to a plant’s root system, has been driven by the development of physical and chemical properties that are the best available. This has been accompanied by progress in plant nutrition, the integration of modern fertilization methods into irrigation techniques, and the introduction of automation technologies. These developments have contributed to the prominence of this cultivation approach and align well with Agriculture 4.0 [93].

A potting medium serves as an optimal environment for plant growth and facilitates the establishment of the root system. The selection of a potting medium has a significant impact on plant growth. Although soil is a common choice due to its ready availability, it presents challenges in large-scale production and is susceptible to soil-borne diseases [15]. In soilless culture systems, lightweight potting media, such as sphagnum moss, rockwool, cocopeat, and perlite, are preferred due to their ease of handling [94]. Such substrates require a continuous supply of water to sustain plant growth. This results in increased operational costs associated with equipment such as pumps and sensors [95]. Hydrogels, which are lightweight, are applied as a conditioning and water-providing component of these substrates [96]. For instance, the incorporation of cellulose hydrogel into the perlite substrate resulted in a 27.9% and 47.7% enhancement in water retention capacity compared to the unamended substrate, with 1% and 2% hydrogel concentrations, respectively. This improvement was achieved while maintaining the high air capacity characteristic of certain granular substrates, such as perlite [97]. 

The utilization of biodegradable hydrogels, incorporating a diverse range of non-toxic cellulose-based materials, such as cellulose derivatives, sawdust cellulose, rice ash, wheat straw, pineapple peel, and oil palm empty fruit bunch, has emerged as a promising avenue in urban farming applications [15,98,99,100].

In recent years, the application of polymer hydrogels as a standalone soilless substrate for plant cultivation, especially for greenhouse vegetables, has been discussed. Figure 2 depicts an example of a hydrogel medium for plant growing, adapted from reference [101]. The authors demonstrated that the cellulose anionic hydrogel was a beneficial medium for seed germination and growth.

When using hydrogel as a soilless medium, a number of factors must be considered.
The hydrogel must possess specific mechanical strength, including the capacity to maintain its own shape and support both the plant roots and aerial parts. Some plants require a certain substrate density for optimal root development (e.g., carrots), making hydrogel substrates unsuitable for such crops.The hydrogel should have good porosity to ensure adequate root zone aeration.The hydrogel should maintain an appropriate moisture level—neither too much nor too little—for the specific plant species. Excessive moisture can lead to mold growth on the hydrogel surface and sprouted seeds.The hydrogel should contain a full spectrum of nutrients necessary for the plant’s balanced development at all stages of its lifecycle and preferably demonstrate antimicrobial activity to protect the plant.The hydrogel should maintain stability throughout the vegetative period of the plant.The hydrogel should possess biodegradability or compostability to facilitate easy disposal after harvest.Ideally, the hydrogel should allow for reusability, such as by reloading with a nutrient solution for subsequent use.

Hydrogels with fillers are capable of meeting these requirements. For example, an agarose-based hydrogel can be filled with a nutrient material consisting of vermiculite, peat, tree bark, and dolomitic lime [95]. The aforementioned mixture was integrated into a hydrogel, which served as a substrate for microgreens, specifically red cabbage, with a growing period of 12 days. The nutrient supplement addressed several issues simultaneously. It increased the overall pore volume and expanded the prevalence of pores > 30 µm by 8 times and provided nourishment and moisture for microgreens throughout the entire development period without additional support. It is also noteworthy that, in addition to agarose, a range of other natural and synthetic polymers were investigated, including gelatin, gellan gum, sodium alginate, locust bean gum, polyacrylamide, sodium polyacrylate, and polyvinyl alcohol. The agarose substrate exhibited the highest germination rate and stem growth. The authors proposed the developed lightweight, convenient, and water-saving hydrogel substrate, which effectively supports plant growth for indoor and space agriculture applications [95]. 

Figure 3 presents photographs of two polymer hydrogels: synthetic polyacrylate (a) and cellulose-based derived from recycled paper (b). It is evident that the acrylate hydrogel, despite its higher water retention capacity, is unsuitable for use as a hydrogel medium for cultivation. Some authors have observed [102] that natural hydrogels exhibit limited toughness and poor mechanical properties when compared to synthetic polymers. It is important to note that the properties of synthetic polymers can vary significantly depending on the monomers and polymerization conditions. With regard to natural polymers, their properties are contingent upon the concentration and the type of cross-linking. Consequently, the mechanical properties of the hydrogel are of paramount importance for its suitability as a germination medium. It is possible to identify suitable hydrogels among both natural and synthetic options.

Among the promising technologies involving hydrogels in urban farming are agro-textiles and 3D printing. For example, vertical substrates, such as knit, woven, and non-woven textiles integrated with hydrogels, have been proposed as soilless agro-textile beds [103]. Additionally, 3D printing technology, also known as additive manufacturing, has been expanding its applications into the agricultural industry [104]. 

In conclusion, hydrogels exhibit considerable potential as a potting medium in agriculture, offering a versatile alternative to traditional soil-based farming methods. Due to their distinctive characteristics, hydrogels have the potential to transform soilless farming practices, offering a sustainable and efficient solution for cultivating crops. As research in this field progresses, hydrogels may emerge as a viable substitute for soil, offering numerous benefits. These include enhanced water retention, improved nutrient delivery, and increased crop yields. By capitalizing on the potential of hydrogels, we can develop novel strategies for agricultural production, laying the foundation for a more resilient and productive farming industry.

## 5. Effects of Polymer Hydrogels on Plant Development

The application of hydrogel additives around a plant seed ensures a prolonged period of water availability, which, in turn, promotes seed growth [97,105]. During the process of imbibition, dry seeds undergo a swelling and shape change due to the rapid initial uptake of water, which is followed by the initiation of embryo expansion [106]. The retained water facilitates the mobilization of storage reserves, subsequently promoting seedling growth. This was demonstrated in the utilization of cellulose hydrogel as a medium for maize seed germination [107]. The authors concluded that hydrogels can act as a substitute for water in the absence of regular irrigation, promoting the germination and growth of maize seeds. Another research group reported that the germination of maize seeds is slightly impeded by the application of a swollen hydrogel (derived from maize straw and acrylic acid) in lieu of the ordinary watering method (Table 1) [105]. The authors posit that this phenomenon may be attributed to the inability of the hydrogel to rapidly release absorbed water in sufficient quantities for the maize seed, preventing it from reaching a saturated state conducive to germination. Consequently, the timing of water release from the hydrogel is of significant importance for seed germination. If the release occurs rapidly, for example within 10–12 h, such hydrogels have the potential to replace the water source, thereby activating the embryo. Hydrogels with a slower release time (several days or weeks) are not suitable for this task.

The continuous presence of moisture can facilitate the acceleration of both root and shoot growth in plants. For instance, Tao et al. observed that the application of the carbohydrate-based hydrogel resulted in a significant increase in the shoot length of maize (3.09 cm) in comparison to the control group (2.06 cm) on the seventh day of the experiment [105]. The formulations of chitosan-based hydrogels led to the growth of plants, with an increase of approximately 70% compared to the reference soil [31]. 

The impact of the synthetic (acrylic acid) hydrogel on maize growth was found to vary with its concentration. In sandy loam, loam, and paddy soil, the addition of 0.2% hydrogel resulted in enhanced seedling and root length. However, the addition of 0.5% did not result in any discernible effect on seedling height, with root growth being inhibited. At a hydrogel content of 1%, both seedling height and root length were negatively impacted [108]. This example demonstrates that excessive moisture does not invariably result in a beneficial outcome. Consequently, the quantity of hydrogel must be tailored to each soil type and potentially to different plant groups, including those that thrive in moisture and those resistant to drought. 

The application of hydrogel in the soil has a significant impact on plant biomass [31]. As indicated in [105], the use of an appropriate amount of hydrogel, as illustrated in the experiments with maize cultivation, has been shown to promote the development of main and lateral roots. Plants with well-developed roots exhibit enhanced nutrient transport, which leads to improved accumulation of both aboveground and belowground biomass. It is evident that an adequate water supply is of paramount importance for the sustenance of plant growth during the fundamental period, as insufficient moisture can impede plant development [109].

In a separate study on maize cultivation under three irrigation levels (adequate, moderate, and deficit), it was demonstrated that the use of hydrogel resulted in a reduction of biomass accumulation by 11.1% under normal irrigation (Table 1). Nevertheless, it significantly enhanced biomass by 39.0% under moderate irrigation and by 98.7% under deficit irrigation [110].

In the continuation of this investigation, it was observed that 8 weeks after sowing, maize height and leaf area experienced significant increases of 41.6% and 79.6%, respectively, under deficit irrigation with hydrogel application compared to the control group without the hydrogel (Table 1). In the context of adequate and moderate irrigation, the hydrogel had a minimal impact on shoot dry mass, yet led to a significant increase of 133.5% and a 27.8% rise in leaf water potential under deficit irrigation. The enhanced growth and water use efficiency of maize in soil with hydrogel under deficit irrigation were attributed to the maintenance of higher relative water contents in leaves, intercellular carbon dioxide concentrations, net photosynthesis, and transpiration rates [111].

The impact of urea-loaded cellulose hydrogel on the growth and yield of upland rice was investigated. The application of a controlled-release fertilizer at the optimal nitrogen rate of 90 kg/ha resulted in the highest grain yield, which increased by 71% compared to the control (Table 1). This was due to an increase in both panicle number and the number of grains per panicle. Furthermore, the application of a controlled-release fertilizer with a moderate nitrogen loading (75% of the total nitrogen applied) yielded the highest crop yield and improved N efficiencies. In conclusion, the application of controlled-release fertilizer had a positive effect on all growth parameters and yield in comparison to the split application of conventional urea fertilizers [112].

Consequently, hydrogels create an environment that is conducive to overall plant vitality. This effect is particularly noticeable in conditions of growth with irrigation deficits. Indeed, in the presence of sufficient irrigation, the plant develops normally in accordance with the natural growth pattern designed by nature. In arid climates where moisture is scarce, the hydrogel will facilitate the creation of adequate humidity for plant development.

**Table 1 gels-10-00368-t001:** Effects of hydrogels on plant development.

Hydrogel Content/ Type	Fertilizer	Seed Germination Rate, %	Root, cm	Shoot, cm/mm	Biomass, g	Crop Yield, g	Ref.
Cellulose nanofibers (NATURAL)	Urea	1 day: 60 vs. 25 (140% profit), 2 days: 78 vs. 52 (50% profit), 3 days: 90 vs. 78 (15% profit), 4 days: 100 vs. 90 (11% profit), 5 days: 100 vs. 100 (no profit)	8 vs. 7 (*p* > 0.05)	2 cm vs. 3 cm (33% loss)	33 vs. 35 (*p* > 0.05)	N/A	[101]
Agarose (NATURAL)	Vermiculite, moss, tree bark, dolomite lime	1 day: 20 vs. 5 (300% profit), 2 days 90 vs. 75 (20% profit)	N/A	Stem length: 17–40 mm, leaf area: 10–40 mm^2^	4.22–9.35	N/A	[95]
Waste paper cellulose, CMC (ARTIFICIAL)	Urea	N/A	N/A	Plant height: 129.6 cm vs. 86.1 cm (51% profit), tillers per plant: 12 vs. 5 (140% profit), leaf chlorophyll content: 39.5 vs. 28 (41% profit)	N/A	1000-grain weight (g): 24.85 vs. 20.46 (21% profit), panicles per pot: 12 vs. 6 (100% profit), grains per panicle: 134 vs. 110 (22% profit), harvest index (%): 45.5 vs. ~37 (23% profit)	[112]
Waste paper cellulose, CMC (ARTIFICIAL)	Urea	53 vs. 27 riversand (96% profit), 18 clayey soil (194% profit), 35 loamy soil (51% profit), daily germination: 1.71 vs. 0.87 riversand (97% profit), 0.58 clayey soil (195% profit), 1.13 loamy soil (51% profit)	N/A	N/A	N/A	N/A	[113]
CMC, HEC (ARTIFICIAL)	N/A	N/A	180 vs. 158 (14% profit)	N/A	1753 vs. 913 (92% profit)	858 vs. 364 (136% profit)	[97]
Super-absorbent polymer (SYNTHETIC)	N/A	N/A	N/A	Plant height and leaf area increased by 41.6 and 79.6% under deficit irrigation	Total: adequate irrigation reduced by 11.1%, moderate irrigation increased by 39.0%, deficit irrigation increased by 98.7% Aboveground: increased by 133.5% under deficit irrigation	N/A	[110,111]
Maize straw, acrylic acid (HYBRID)	N/A	No toxic effect on the seeds	4.93 vs. 2.31 (113% profit)	3.09 cm vs. 2.06 cm (50% profit)	Total: 0.15 vs. 0.05 (200% profit), aboveground: 0.11 vs. 0.03 (267% profit)	N/A	[105]
Poly(acrylamide), CMC (HYBRID)	MMt	86.85 vs. 89.26 (*p* > 0.05)	Root superficial area (mm^2^): 35.91 vs. 19.58 (83% profit), root volume (mm³): 0.724 vs. 0.267 (171% profit)	Shoot diameter (mm): 2.22 vs. 2.03 (9% profit)	Aboveground dry: 0.108 vs. 0.095 (*p* > 0.05)	N/A	[100]

The results are provided as the best of the series of experiments vs. control sample; profit (P, %) was calculated by the authors as $P=\frac{treated-control}{control}\cdot 100\%$. If there was no significant difference between the results, *p* > 0.05 was noted. CMC—carboxymethyl cellulose, HEC—hydroxyethylcellulose, MMt—calcium montmorillonite.

## 6. Natural vs. Synthetic Hydrogels: Which Are Better?

A considerable number of reviews and book chapters have been written on the classification of polymer hydrogels [114,115,116] and, thus, the current aim is not to discuss the full range of features used to classify polymers once more. The focus of this study is on the aspects that have a general impact on the applicability of hydrogels in agriculture. It is postulated that the primary characteristics of hydrogels are largely dependent on their origin. These include price, liquid retention capacity, sorption ability, mechanical properties, and, of course, carbon footprint. Some researchers have emphasized the importance of natural polymers as renewable and biodegradable [11], while others have advocated for synthetic polymers due to their outstanding properties and low price [19] or hybrid hydrogels that gather all the benefits of both classes [12]. These characteristics are somewhat disparate and in order to gain a comprehensive understanding, the current review examines the entire life cycle of the hydrogel.

Figure 4 depicts the five stages of the product life cycle, with a classification of hydrogel material types relevant to each stage. A brief description of the principal processes occurring at each stage for each type of hydrogel is provided, along with an indication of their positive and negative key characteristics. This study does not include a comprehensive analysis of the oil refining process or a stage-by-stage examination of the numerous methods of hydrogel production. The authors posit that the succinct description of processes and characteristics presented is sufficient for general assessment.

Hydrogels can be classified in a manner analogous to polymers, based on their initial materials, as follows: synthetic (petroleum-based), natural (biomass-derived), artificial (derived from chemically modified natural polymers), and hybrid/semi-synthetic (obtained by cross-linking natural and synthetic polymers).

Biomass-derived hydrogels are derived from a variety of sources, including (1) plants and algae (cellulose, lignin, starch, alginate, agarose); (2) food, agricultural, wood, and paper wastes (cellulose, starch); and (3) animals, insects, and microorganisms (chitosan, collagen, gelatin). These polymers are biosynthesized by nature, which is why the second stage of the lifecycle—the production of precursors—is absent for them (Figure 4, stage 2). Nevertheless, some natural polymers can undergo processing to obtain enhanced forms. For instance, cellulose can be transformed into microcrystalline or nanocellulose through chemical or fermentative hydrolysis [117]. In addition to the aforementioned forms, natural polymers are frequently subjected to derivatization in order to overcome limitations, such as the poor solubility of cellulose. This is achieved by increasing the number of reactive functional groups, which subsequently helps to retain active ingredients in the hydrogel. 

Examples of artificial polymers include carboxymethyl cellulose, hydroxypropyl methylcellulose, carboxymethyl chitosan, and N,N,N-trimethylchitosan. Derivatization confers advantages in terms of the diversity of properties but also contributes to an increase in the product’s carbon footprint.

The source material for synthetic hydrogels typically involves the use of significant quantities of petroleum, which are refined into monomers in the second stage of the product’s life cycle. These monomers include ethylene glycol, acrylic acid, acrylamide, vinyl alcohol, and others, which are part of the second stage of the life cycle [20]. It is evident that in the first and second stages of the lifecycle, which relate to the extraction and processing of raw materials, biomass-derived polymers have a significant advantage (Figure 4). The disadvantages of the technology, including low conversion rates, numerous harmful emissions and effluents, and water and solvent consumption, can be mitigated by employing the best available technologies.

Moreover, this phenomenon is intrinsic to both synthetic and natural polymers. However, the most significant challenge, which does not have an easy solution, is that the sole source of obtaining synthetic hydrogels is petroleum, and its reserves are finite.

The production stage of hydrogels (Figure 4, stage 3) is dependent on the type of precursor, with polymerization or dissolution processes occurring accordingly. Subsequently, the gelation process is initiated through physical or chemical cross-linking. Among synthetic hydrogels for agricultural applications, the following can be mentioned: poly(vinyl alcohol) [67], poly(urethane) [118], poly(2-hydroxyethyl methacrylate) [20], poly(acrylic acid) [119], and poly(acrylamide) [120], while cellulose [113], nanocellulose [121], chitosan [31], chitosan/casein [122], alginate [123], starch [124], and lignin [125] hydrogels can be considered natural ones. An example of artificial hydrogels is sodium carboxymethyl cellulose [126]. At this stage of the hydrogel’s life cycle, a new class of hydrogels emerges, created through the cross-linking of natural and synthetic polymers. Some examples of these include (acrylamide, acrylic acid)/starch/poly(ethylene glycol) [127], gellan gum/poly(ethylene glycol) diacrylate [128], and poly(acrylamide)/cashew tree gum [129] hydrogels. 

The level of technological readiness (TRL) plays a significant role in the third stage of the product lifecycle. The polymerization of synthetic polymers is carried out on an industrial scale (TRL 9) and the technology is optimized to minimize business costs. It is unfortunate that the interests of business do not always align with those of environmental preservation. Nevertheless, at the level of environmental legislation, the basic requirements for pollution prevention are adhered to. Conversely, natural and synthetic polymers, as well as emerging hybrid polymers at this stage of the lifecycle, are predominantly at lower levels of technological readiness. At best, this involves pilot-scale production (TRL 6), and, more often, only laboratory-level experiments (TRL 3–4). There is a substantial amount of work to be done to transfer these developments to the industrial scale. This is a significant challenge, but one that can be overcome with the support of the government and major businesses.

At the stage of using hydrogels as reservoirs for water and/or solutions for the targeted delivery of active ingredients (Figure 4, stage 4), it is challenging to identify a preferred type of hydrogel. All of the aforementioned hydrogels can fulfill agricultural tasks, although with varying efficiency. For instance, the water retention capacity of synthetic and hybrid hydrogels can exceed that of natural and artificial ones by a factor of several orders of magnitude. However, the latter has other advantages in terms of usage, namely, they possess greater mechanical strength, which is beneficial, for instance, when using soilless substrates in urban farming, where the hydrogel must retain seeds within its volume and serve as a framework for future seedlings. Consequently, it can be concluded that, depending on the specific requirements, an appropriate hydrogel can be selected. At this stage, no clear advantage can be attributed to any specific type of hydrogel.

The final stage of the hydrogel lifecycle (Figure 4, stage 5) concerns the behavior of the hydrogel following its use. In this context, the capacity of the hydrogel to undergo biodegradation is of paramount importance. The majority of synthetic polymers are not biodegradable. Consequently, they remain in the soil for many decades or even centuries, gradually breaking down into microplastics [11] and potentially releasing harmful components that, when entering groundwater, will have a negative impact on water bodies. In contrast, natural and artificial polymers undergo biodegradation rapidly (sometimes too rapidly) by numerous species of bacteria and fungi present in the soil as part of the evolutionary process of processing plant and animal residues, which are the basis of natural polymers. The biodegradability of hybrid polymers is contingent upon their quantitative and qualitative composition.

In conclusion, an analysis of the entire lifecycle of hydrogels for agriculture reveals that, despite the advantages of synthetic hydrogels (low cost, outstanding water retention properties, etc.), their use in Agriculture 4.0 is undesirable due to the non-renewability of resources and disposal challenges. The remaining three types of hydrogels, namely, natural, artificial, and hybrid, hold considerable potential for use but require significant efforts from scientists and technologists on the one hand and government and businesses on the other to achieve industrial implementation. Once the technology for producing hydrogels has reached TRL 9, issues with the recovery and re-direction of reagents and by-products will be resolved, and the target product will be produced in industrial quantities. This will result in a significant reduction in the cost of production.

## 7. Conclusions

In conclusion, the distinctive characteristics of polymer hydrogels facilitate water efficiency, soil enhancement, and the targeted delivery of essential components, rendering them invaluable tools in modern agriculture with applications spanning from traditional farming to innovative soilless cultivation methods. Hydrogels facilitate various aspects of plant development, including seedling survival, germination rate, and plant and root growth while also preventing soil erosion. Furthermore, their capacity to facilitate a sustained release of fertilizers serves to mitigate the risk of chemical overdosing of the soil and water bodies. The use of hydrogels carrying fertilizers or pesticides enables a reduction in the environmental impact associated with agrochemicals, minimizing losses arising from leaching, volatilization, and degradation while simultaneously preserving the effectiveness of these chemicals. Smart hydrogels exhibit responsiveness to variations in pH, temperature, light, ionic strength, osmotic pressure, or changes in magnetic or electric fields, enabling precise control over the rate of release. The combined properties of polymer hydrogels position them as a promising contributor to the evolution of agriculture in the context of Agriculture 4.0.

It is recommended that active ingredients be delivered using bio-based hydrogels due to their high biodegradation rate, which is facilitated by soil microbial consortia. Once the carrier has fulfilled its function by releasing nutrients or protective substances, its residues should be rapidly utilized by bacteria and fungi, as a new batch of active ingredient-loaded hydrogels will be introduced into the soil during the subsequent cycle. Over time, if the degradation rate is low (as observed with acrylates, which exhibited a mere 2% degradation per year), the natural composition of the soil will be rapidly disrupted. Consequently, a degradation period comparable to the plant’s growing season appears to be optimal for serving as an active ingredient delivery system.

For hydrogels incorporated as soil conditioners, a longer service life is preferable. In this scenario, the primary objective is to ensure that the hydrogel’s effective working period does not fall below the plant’s lifespan. In the context of orchard soil conditioning, for instance, hybrid hydrogels with a degradation period spanning several years would be advantageous. Conversely, for the cultivation of cereals, hydrogels with a degradation period of one year would be more suitable. For the cultivation of greens with a vegetative period of a few months, hydrogels derived from non-derivatized natural polymers may be a suitable option.

## Figures and Tables

**Figure 1 gels-10-00368-f001:**
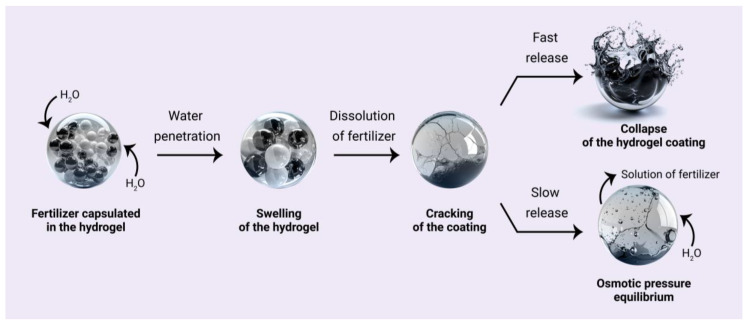
The mechanisms of nutrient release from the hydrogel can be described as follows: high osmotic pressure results in the collapse of the coating, leading to a sudden release (the first outcome); gradual release of the fertilizer solution due to balancing the difference in osmotic pressure inside and outside the core (the second outcome).

**Figure 2 gels-10-00368-f002:**
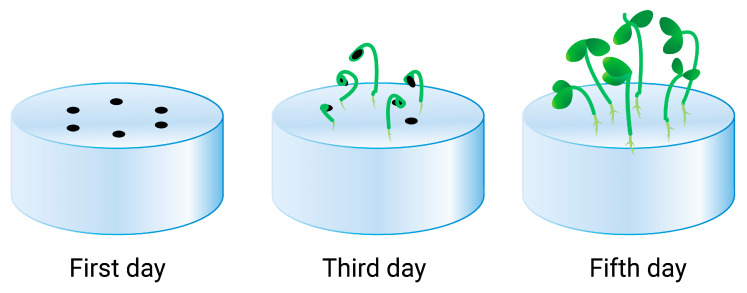
Seeds planted on the surface of the hydrogel substrate after 1, 3, and 5 days.

**Figure 3 gels-10-00368-f003:**
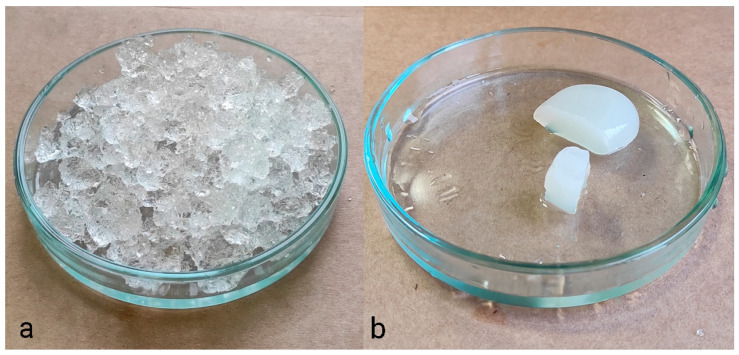
Hydrogel based on polyacrylate (**a**) and cellulose-based hydrogel (**b**).

**Figure 4 gels-10-00368-f004:**
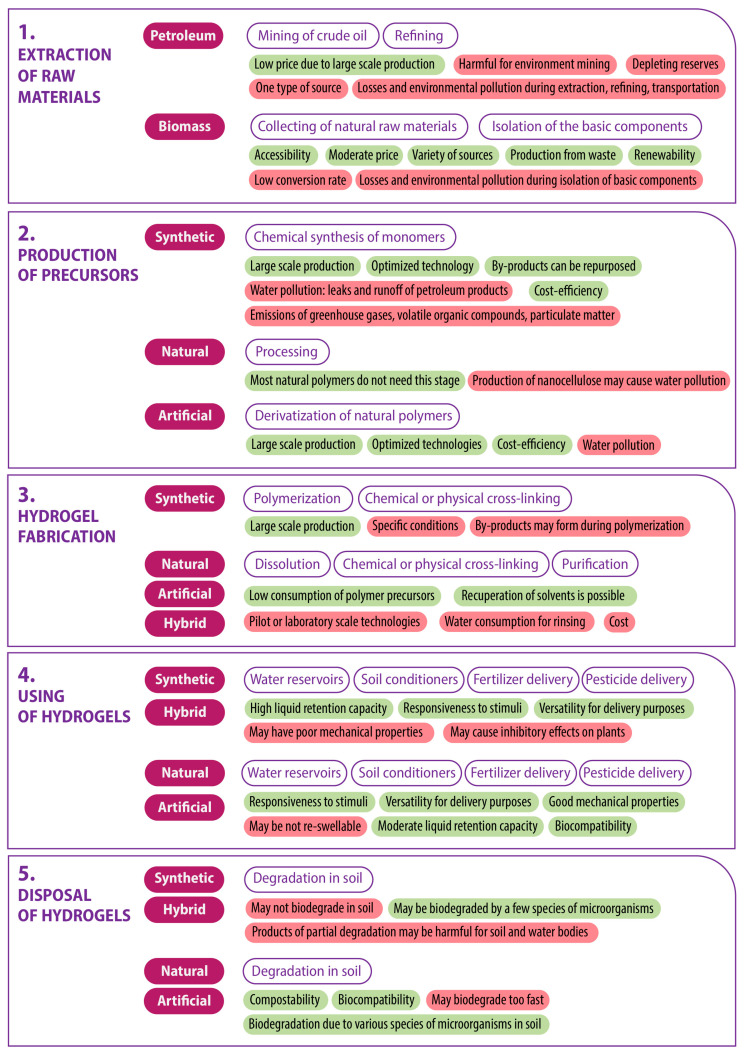
Life cycle assessment of the polymer hydrogels for agricultural applications. Key characteristics of the processes at each stage are colored in green if positive and red if negative.

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
