# Peer review of "Agriculture 4.0: Polymer Hydrogels as Delivery Agents of Active Ingredients"

_gels, 2024, doi:10.3390/gels10060368_

Round 1
Reviewer 1 Report
Comments and Suggestions for Authors
In these review, the authors reviewed that, 1) polymer hydrogels have become a promising material for enhancing agricultural productivity due to the ability to retain and then release water, which can help alleviate the need for frequent irrigation in dryland environments. 2) the controlled release of fertilizers by the hydrogels decreases chemical overdosing risks and the environmental impact associated with the use of agrochemicals. 3) The potential of polymer hydrogels in sustainable agriculture and farming, identifying their impact on soil quality, is depicted with the focus on the ability of delivering nutritional and protective active ingredients. 4) The impact of hydrogels on plant growth, development, and yield was discussed. The authors regarded as, the question of which hydrogels are more suitable for agriculture − natural or synthetic − is debatable, as both have their merits and drawbacks. An analysis of polymer hydrogels life cycle in terms of their initial material has shown the advantage of bio-based hydrogels, such as cellulose, lignin, starch, alginate, chitosan and their derivatives and hybrids, that aligns with sustainable practices, reducing dependence on non-renewable resources.
1) The newly-developed concept and academic opinions are attractive. The figures are more intuitive and make one think. I will recommend this paper
2) This study gave focus on the contribution of polymer hydrogels on soil nutrition and plant growth. In the year of 2050, the world population will be 9 billion, the contribution of polymer hydrogels to seed germination rate, crop production and arable land area had better be predicted.
3) In the stage of crop or other plants production, which stage will be possible for using agriculture 4.0?A typical use scenarios had better be given.
Reviewer 2 Report
Comments and Suggestions for Authors
Please, find attached below few remarks.

Reviewer 3 Report
Comments and Suggestions for Authors
The paper provides a compelling exploration of the progress made in utilizing hydrogels in agricultural contexts, showcasing notable achievements thus far. Considering the extensive dataset presented, incorporating additional tables would enhance clarity for readers. Additionally, it is imperative to address the growing concern of micro and nanoplastics by conducting a thorough literature review on the subject. The comments are in the paper.
